# IGFBP-6: At the Crossroads of Immunity, Tissue Repair and Fibrosis

**DOI:** 10.3390/ijms23084358

**Published:** 2022-04-14

**Authors:** Arcangelo Liso, Santina Venuto, Anna Rita Daniela Coda, Cesarina Giallongo, Giuseppe Alberto Palumbo, Daniele Tibullo

**Affiliations:** 1Department of Medical and Surgical Sciences, University of Foggia, 71100 Foggia, Italy; santina.venuto@unifg.it (S.V.); daniela.coda@unifg.it (A.R.D.C.); 2Department of Medical Surgical Sciences and Advanced Technologies “G.F. Ingrassia”, University of Catania, 95123 Catania, Italy; cesarina.giallongo@unict.it (C.G.); palumbo.gam@gmail.com (G.A.P.); 3Department of Biomedical and Biotechnological Sciences, University of Catania, 95123 Catania, Italy; d.tibullo@unict.it

**Keywords:** IGFBP-6, fibrosis, immunity, CAFs, IGFBPs

## Abstract

Insulin-like growth factors binding protein-6 (IGFBP-6) is involved in a relevant number of cellular activities and represents an important factor in the immune response, particularly in human dendritic cells (DCs). Over the past several years, significant insights into the IGF-independent effects of IGFBP-6 were discovered, such as the induction of chemotaxis, capacity to increase oxidative burst and neutrophils degranulation, ability to induce metabolic changes in DCs, and, more recently, the regulation of the Sonic Hedgehog (SHH) signaling pathway during fibrosis. IGFBP-6 has been implicated in different human diseases, and it plays a rather controversial role in the biology of tumors. Notably, well established relationships between immunity, stroma activity, and fibrosis are prognostic and predictive of response to cancer immunotherapy. This review aims at describing the current understanding of mechanisms that link IGFBP-6 and fibrosis development and at highlighting the multiple roles of IGFBP-6 to provide an insight into evolutionarily conserved mechanisms that can be relevant for inflammation, tumor immunity, and immunological diseases.

## 1. Introduction: The Insulin-Like Growth Factors-Binding Proteins

The insulin-like growth factors (IGFs) are peptides sharing significant structural homology with insulin, which were mostly investigated at the end of the last century [1]. Circulating IGFs are complexed to a family of structurally related binding proteins named IGF-binding proteins (IGFBPs). Although the existence of IGFBPs was suspected quite a long time ago, they were cloned and sequenced in the mid-1980s to early 1990s. IGFs, including IGF-I and IGF-II, are members of the insulin superfamily of growth promoting peptides and are finely regulated by IGFBPs, a class of secreted proteins exerting multiple functions in different tissues, prolonging the half-life and also modulating the availability of circulating IGFs [2].

Importantly, although members of the IGFBPs family share significant sequence homology, each of them has some unique structural features, playing distinct roles in many cellular processes [3]. IGFBPs genes also have different regulation manners and distinct expression patterns. In spite of this functional and regulatory diversity, it has been puzzling that loss-of-function studies have yielded relatively little information about the physiological functions of IGFBPs. In fact, it has been proposed that evolution led to retain so many IGFBPs in order to facilitate the fine-tuning of IGFs signaling. However, another interesting, proposed explanation is that many IGFBPs functions have evolved to allow adjustment of IGFs signaling under stressful conditions, which would likely not be revealed in standard laboratory settings.

IGFBPs have more recently been found to bind to their own receptors or to translocate into the interior compartments of cells where they may execute IGF-independent actions. Notably, Bach L.A. previously reviewed in detail why IGFBP-6 appears to be different from other IGFBPs both in structure and in function as it binds IGF-II with great affinity [4], being a relatively specific inhibitor of IGF-II actions. Indeed, IGFBP-6 inhibits cellular differentiation [5].

Importantly, a number of mechanisms are involved in this phenomenon, including the modulation of other growth factor pathways, nuclear localization and consequent transcriptional regulation, interference with the sphingolipid pathway, binding to non-IGF biomolecules in the intra- and extracellular space, and on the cell surface too. Importantly, most IGFBPs finely regulate a lot of essential biological processes important for cell proliferation and cell cycle progression. Their functions are properly related to cellular growth and differentiation as well as immune regulation [6]. Further studies are warranted to improve our understanding of IGFBPs biology and broaden define their cellular roles and determine their therapeutic potential.

In light of many recent studies on IGFBP-6 function, here, we propose that IGFBP-6 may play a role in tissue remodeling, fibrosis, and immunity starting with highlighting its possible role in the immunological response and in tissue repair.

## 2. Evolution, Immunity, and Tissue Repair

The immune system has evolved not only as defense machinery, but also as a player in the maintenance of tissue integrity [7]. In fact, the collateral tissue damage after a massive pathogen attack could seriously compromise the host fitness. Thus, the most gainful approach is to favor the tissue repair process.

Indeed, Th1 immunity, evolved to kill intracellular parasites and to produce a pro-inflammatory response, while Th2 cells evolved to cope with larger extracellular organisms and to provide a repair response to tissue destructive pathogens (e.g., helminths) [7,8]. Notably, when human cytokines activities fall into the Th2 pattern, promote the formation of granuloma and matrix deposition, which naturally evolve in open lesions. Then, Th2 specific memory and adaptative immunity are both relevant to tolerate mechanisms that minimize host damage accelerating wound resolution, and consequently reducing unfavorable effects on secondary infection (e.g., hemorrhage due to lung-migrating nematodes) [7] exposures. In this response, another relevant matter is the time, as tissue takes a significant amount of time to return to its original architecture [7]. Conversely, the rapid primary response to damage results in a granulation tissue, preventing bacteria invasion. Granulation tissue is a stromal fibrotic framework that replaces a fibrin clot in a healing wound, histologically characterized by the presence and proliferation of fibroblasts, keratinocytes, endothelial cells, extracellular matrix inflammatory infiltration, and new thin-walled capillaries. The most extensive evidence that Th2 is involved in tissue repair is properly that Interleukin-13 (IL-13), a potent pro-fibrotic cytokine, plays a critical role as Th2 specific ligand. Besides Regulatory T-cells, Th2-activated M2 macrophages also participate in wound resolution while at the same time regulating matrix turnover [9]. This process ends with efficient wound closure and a consequently total repair until the complete shutdown of the inflammatory response. Thus, pro-regenerative tissue repair and anti-inflammatory functions are the most relevant features of Th2 immunity. Interestingly, the increased expression of Th2 and its related cytokines, lead to the secretion of several chemokine ligands from epithelial cells and fibroblasts that, in response to IL-13 signaling, recruit eosinophils [9], one of the cell types in which IGFBP-6 is most expressed [10].

Consistent with this, Th2-activated macrophage products can play different roles depending upon the activation of several spatiotemporally orchestrated pathways [11]. Importantly, macrophages can switch from pro-inflammatory to a growth-promoting, reparative phenotype or to a mixed profile linked to anti-inflammatory and wound healing functions.

In this contest, Transforming Growth Factor-β (TGF-β) can be a potent pro-fibrotic mediator, suppressing the pro-inflammatory responses. TGF-β and IGFBP-6 have a complex relationship. IGFBP-6 expression has previously been reported to be stimulated by Interleukin-1b and tumor necrosis factor-alpha but inhibited by the presence of TGF-β [12,13]. Indeed, TGF-β inhibits IGFBP-6 expression in osteoblast-enriched cells from fetal rat calvariae, by transcriptional mechanisms [14]. Moreover, transfection, together with treatment of desmoids with TGF-β, revealed that IGFBP-6 is a target of TGF-β signaling in desmoids independently of β- Catenin/T-cell factor signaling [15]. Other studies have to be performed to answer these questions in desmoids. TGF-β stimulation alters the transcriptional activity of IGFBP-6, a β-Catenin responsive promoter, also determining fibroblasts tumoral phenotype.

Fibroblasts are a cell population responsible for tissues and organs structural framework, by remodeling extracellular matrix (ECM) proteins and supporting homeostasis. They have key roles in many conditions, such as fibrosis, cancer, autoimmunity, and wound healing. Buechler M.B. and colleagues recently showed transcriptional similarities between murine and human fibroblasts, suggesting that mouse fibroblasts constitute a possibility to understand fibroblast subtypes in human disease states [16].

IGFBP-6, therefore, plays a role in both the conduction of the immune response and probably in the repair after tissue damage. In the next paragraphs, we will discuss how IGFBP-6 regulation is modified during the development of the immune response and subsequently in the control and progression of the fibrotic mechanisms.

## 3. IGFBP-6 Has an Important Role in the Immune Response

We have recently demonstrated that IGFBP-6 plays a putative role in the immune system, inducing chemotaxis of monocytes and T-lymphocytes and playing a functional role in the hyperthermic response [17]. In fact, previous work has shown that exposure to hyperthermia significantly impacts the immunostimulatory capacity of DCs, inducing their specific genetic and metabolic reprogramming [18]. It is known that stressful conditions can highlight novel aspects of biology and immunology. Starting from the observation that normal donor’s DCs upregulate gene transcription in response to hyperthermia, more recently we demonstrated that human monocyte-DCs subjected to hyperthermia show a distinct gene expression profile with selective upregulation of IGFBP-6 [17,19]. We also demonstrated and described several previously unknown functions of the protein, like the ability to increase oxidative burst and degranulation of neutrophils. IGFBP-6 is also overexpressed in the serum and the joints of rheumatoid arthritis patients and is able to induce great in vitro T-lymphocytes migration [20].

Intriguingly, IGFBP-6 is also an acute-phase protein, as demonstrated by our and other groups in experiments showing that it is rapidly produced in response to damage after DCs and fibroblast exposure to H_2_O_2_ [21]. Particularly, H_2_O_2_ induces a dose-dependent up-regulation of IGFBP-6 mRNA and protein levels in skin diploid fibroblasts exposed to a sublethal dose of H_2_O_2_ [22]. It has also been reported that hypoxia induces *IGFBP-6* upregulation in endothelial cells [23]. IGFBP-6 is also secreted by human bone marrow-derived mesenchymal stem cells (hMSCs), multipotent cells that make the tissue microenvironment more favorable for tissue repair by the secretion of different growth factors. hMSCs secreted IGFBP-6 has a protective effect on H_2_O_2_-injured primary cortical neuron cultures [24].

Oxidative stress in the central nervous system generates reactive oxygen species that contribute to the pathogenesis of several neurodegenerative diseases [25]. In this context, IGFBP-6 is an agonist of neutrophils’ functions like increased oxidative burst with reactive oxygen species (ROS) production, degranulation of primary granules, chemotaxis of T-cells and monocytes through the epithelial monolayer [26]. IGFBP-6 was also found as the most abundant secretory protein in the conditioned medium of cells that improved the symptoms of Parkinson’s disease [27], and it also belongs to 18 signaling secreted proteins that can be used to classify Alzheimer’s samples [28]. IGFBP-6 is a secreted protein that performs various immunological functions. Although several studies have highlighted its chemotactic and pro-inflammatory role, recent studies have highlighted a possible new role of IGFBP-6 which would seem not to have an exclusively pro-inflammatory but also anti-inflammatory function. Indeed, it was demonstrated that IGFBP-6 improves mitochondrial fitness and redox, reducing mitochondrial ROS production and modulating lactate metabolism and oxidative stress in a human breast cancer cell line [29].

There is also other evidence pointing towards the role of IGFBP-6 in immunity. *IGFBP-6* RNA is highly expressed in eosinophils [10,30] and *IGFBP-6* gene has been linked to allergic asthma [31,32,33] the latter suggesting it may play a role in Th2 response, favoring an imbalance of the immune response towards immunosuppressive stimuli. More evidence towards the role of the protein in the control of immunity is presented in the work of Park J.H. et al. that showed IGFBP-6 involvement in thymic atrophy [34].

IGFBP-6 is also highlighted as a gene significantly associated with cytotoxic T cell dysfunction in an association study between distinct stromal gene signatures and immune content in three large independent triple-negative breast cancer patient cohorts. IGFBP-6 also appears to be up-regulated in inflammatory-cancer activated fibroblasts (iCAFs) that are important for maintaining immune suppression and chemoresistance associated with T-cell dysfunction [35].

Finally, IGFBP-6 seems to be linked to the immunology and pathogenesis of bovine paratuberculosis, a chronic enteropathy of ruminants, controlling cellular proliferation in paucibacillary tissues [36]. Particularly, IGFBP-6 was found up-regulated in cattle peripheral blood mononuclear cells (PBMC) stimulated by Mycobacterium avium paratuberculosis [37]. Microarray and real-time RT-qPCR analyses revealed increased IGFBP-6 levels in paucibacillary diseased ileum from affected sheep [36].

In summary, several pieces of evidence suggest an important role for IGFBP-6 in the immune response. IGFBP-6 stimulates chemotaxis and determines, along with other factors, the activation of the immune response. Notably, inflammation and tissue damage are important triggers for regeneration and fibrosis. Tissue damage determines type and polarization of inflammation by recruiting and activating a variety of different cells types of the innate and adaptive immune system [38]. Starting from this point, the following paragraph focuses on IGFBP-6 role in different kinds of fibrosis progression.

## 4. IGFBPs Are Involved in the Regulation of Connective Tissues and of Fibrosis

Human fibrotic diseases share common features of a progressive and deregulated accumulation of fibrotic tissue in affected organs, causing their dysfunction and ultimately failure. The striking heterogeneity in their etiology and clinical manifestations, the absence of suitable and validated biomarkers together with the current absence of therapeutic agents, make these pathologies a very important object of study [39].

During the fibrosis pathogenesis, a number of growth factors, chemokines, and cytokines act together to promote a fibrotic microenvironment, leading to the development of a profibrotic fibroblasts population. Several studies demonstrate that IGFBPs participate in fibrosis progression and could be employed as circulating novel protein panels for diagnosis and as possible therapeutic targets.

### 4.1. IGFBP-6 Regulates Several Fibrosis Mechanisms

There is significant evidence demonstrating the role of IGFBP-6 in fibrotic and connective tissue, also starting with the point that both TGF-β and oxidative stress conditions increase IGFBP-6 expression levels in fibroblasts [22,40,41].

IGFBP-6 is highly expressed in fibroblasts [10,30] and is also involved in the regulation of connective tissues maintenance. IGFBP-6 is expressed and governs the homeostasis and differentiation of periodontal ligament cells besides being the most abundant growth factor expressed by adipose tissue-derived multi-lineage progenitor cells [42,43]. Recently the new IGFBP-6 variant T430C was identified as causative of disc degeneration, a pathological process leading to spinal deterioration [44]. Moreover, in a proteomic study that compared osteoblastic secretome from the sclerotic or non-sclerotic area of subchondral bone osteoarthritis, IGFBP-6 was identified among proteins that were significantly more secreted by sclerotic osteoblasts [45]. Moreover, IGFBP-6 was recently listed as a biomarker of the outer ring fibrosis of intervertebral discs [46].

Several works demonstrated that IGFBP-6 is differentially expressed in dermal, renal, hepatic, cardiac fibrosis, and myelofibrosis (Figure 1 and Table 1).

#### 4.1.1. Dermal Fibrosis

Systemic sclerosis is a chronic, multisystem, autoimmune tissue disorder characterized by tissue visceral fibrosis of the skin and internal organs. It is characterized by vascular dysfunction and immunologic activation leading to an excessive accumulation of ECM in lesioned tissues [49]. Critical signaling cascades, initiated primarily by TGF-β, but also involving numerous cytokines and signaling molecules which stimulate profibrotic reactions in myofibroblasts, offer potential therapeutic targets [39].

Dermal fibroblasts treated with a combination of TGF-β1 and the monocyte chemoattractant protein 3 (MCP-3), a protein that is up-regulated in fibrosis, led to a high and significant IGFBP-6 upregulation [39]. Moreover, in a gene-expression profile study on dermal fibroblasts from type 1 tight-skin mice, IGFBP-6 resulted in upregulation in relation to the expression and potential fibrotic activity of the MCP-3 [49].

Among dermal fibrotic diseases, Dupuytren’s Disease (DD), a common and heritable fibrosis of the palmar fascia, induces the development of hyper-contractile fibroblasts and typically manifests as permanent finger contractures. IGFBP-6 is involved in this fibrotic disease upon the regulation of cellular contractility and proliferation. In particular, IGFBP-6 has a suppressive role by inhibiting the proliferation of primary cells derived from contracture tissues (DD cells), and IGF-II is an inducer of cellular contractility in this connective tissue disease. Downregulated IGFBP-6 and upregulated IGF-II levels also contribute to DD progression [50].

#### 4.1.2. Renal Fibrosis

Several works demonstrate a relation between IGFBP-6 and renal fibrosis. It is well known that IGFBP-6 mRNA and protein expression levels are high in the kidney and, of interest, they are abundant in plasma of adults and children with Chronic Kidney Disease (CKD) or End Stage Renal Disease (ERSD) [51,52], suggesting that IGFBP-6 might be related to the kidney development process. Wang S. and colleagues recently speculated that IGFBP-6 might be involved in the development of kidney fibrosis by regulating apoptosis of renal cells [2]. Among kidneys diseases, a congenital obstructive uropathy is especially characterized by interstitial fibrosis that is a representative sign of renal damage, following renal obstruction. In this context, IGFBP-6 was shown to be significantly up-regulated in the obstructed kidneys of an animal model for congenital obstructive uropathy [53].

#### 4.1.3. Hepatic Fibrosis

Hepatic stellate cells (HSCs) represent the cell type primarily involved in the progression of liver fibrosis and, in their activated phenotype, constitutively produce high quantities of IGFBPs. Interestingly, HSCs isolated from the human liver express high levels of *IGFBP-6* mRNA, which is differentially regulated by IGF-I and TGF-β [59].

Among the most involved cytokines in fibrosis pathogenesis, TGF-β is known to be a predominant mediator of the development of interstitial fibrosis in renal obstruction [60]. An important regulatory loop between chemokines and TGF-β drives the inflammatory and fibrotic diseases. In this context, *IGFBP-6* transcript is regulated by TGF-β, a potent profibrotic cytokine that is critical in the development of fibrotic microenvironment [40]. IGFBP-6 is significantly positively associated with steatosis, and it was recently reported as a potential contributor of hepatic inflammation and fibrosis [54]. A correlation study of IGFBPs with the different stages of fibrosis in Chronic Hepatitis C (CHC) highlights that IGFBP-6 was the protein that was most involved in fibrosis stage identification. Indeed, IGFBP-6 expression was lower in the patients compared to healthy individuals. Moreover, IGFBP-6 was downregulated in the senescence of human fibroblasts, suggesting that it may participate in the regulation of the senescence process and the extracellular matrix deposition, during liver damage from CHC [55].

Likewise, IGF-1 and IGFBPs are involved in the pathophysiology of Non-Alcoholic Fatty Liver Disease (NAFLD) and in the regulation of glucose homeostasis. In the liver, the decrease of IGF-1 levels contributes to the development of NAFLD and non-alcoholic steatohepatitis, also related to the hepatic expression of several IGFBPs. Specifically, IGFBP-6 levels are higher in relation to increasing steatosis, while the hepatic expression of IGFBP-6 is strongly positively associated with fibrosis, steatosis grade, and NAFLD activity score [54]. Remarkably, circulating IGFBP-6 resulted significantly reduced after the treatment with the GHRH-analogue Tesamorelin, which reduces liver fat and prevents fibrosis in patients with NAFLD [61].

#### 4.1.4. Cardiac Fibrosis

Late adverse cardiac remodeling is a sophisticated structural and functional response of the failing heart to numerous triggers, including fibrosis [62]. Interestingly, IGFBP-6 is involved in both Acute Myocardial Infarction (AMI) and in carotid atherosclerotic plaques, both characterized by the development of a fibrotic process. IGFBP-6 co-localizes with CD31^+^ endothelial cells and with CD68^+^ macrophages in the fibrotic cup of the atherosclerotic plaques [57]. Notably, it was recently counted as a central biomarker for the prediction of vulnerable plaques and AMI, as it is markedly downregulated in both unstable human carotid plaques in plasma samples of AMI patients, compared to the controls [56,57].

#### 4.1.5. Myelofibrosis

Many hematologic and non-hematologic disorders are associated with increased bone marrow fibrosis. Among them, primary myelofibrosis (PMF) is a hematologic disease characterized by the progressive proliferation of mainly granulocytic and megakaryocytic cells in the bone marrow [63,64,65].

Recently, we highlighted the emerging role of IGFBP-6 in controlling the fibrotic process implicated in the pathogenesis of PMF patients. Starting with the evidence that IGFBP-6 levels are significantly increased in PMF patients with wild-type Janus Kinase 2, our recent study demonstrated a new role of the axis IGFBP-6/SHH/Toll-like receptor 4 as involved in alterations of the primary myelofibrosis microenvironment and that IGFBP-6 may play a fundamental role in activating SHH pathway during the fibrotic process [58].

As widely discussed in this paragraph, aberrant IGFBP-6 signaling leads to fibrosis in different types of tissues. Notably, unchecked pro-fibrotic and pro-inflammatory signaling can became the starting point for tumor-associated fibrosis [66]. The fibrosis components like cancer-associated fibroblasts (CAFs), extracellular matrix rigidity, and dense collagen deposition are essential regulators of tumor progression but may also be critical mechanisms of immune surveillance. Therefore, in the next paragraph, we will discuss IGFBP-6 signaling in the tumor microenvironment (TME), which harbors cancer-associated fibroblasts, leading to angiogenesis, fibrosis, and immune evasion.

## 5. IGFBP-6 Controls Fibroblasts and TME during Cancer Progression

Immune cells within the TME play an important role in tumorigenesis. Cancer cells interact closely with ECM and stromal cells and form the principal structure of the TME. Within the TME, a heterogeneous population including immune and non-immune cells, cancer cells, and stromal cells (e.g., fibroblasts) is present [67]. Fibroblasts are among the most abundant stromal cells in TME, progressively differentiating into activated, motile, myofibroblast-like, and pro-tumorigenic cells referred to as CAFs [68]. CAFs represent the principal component of the tumor stroma. They are an important target for enhancing cancer immunotherapy, being both a physical barrier and a source of immunosuppressive molecules [69]. In the TME there are numerous types of cells that accumulate and reach the tumor at different stages, and together with CAFs there are also infiltrating inflammatory cells, endothelial progenitors, and bone marrow-derived hematopoietic cells. There is a strong interaction between the microenvironment and the tumor, associated with further events and strategies [67]. Moreover, the TME is fundamental for clinical results and responses to therapy. Tumor-infiltrating immune cells are able to regulate both cancer progression and the effectiveness of anti-cancer therapies, by exerting pro- and anti-cancer actions [70].

Heterotypic interactions between stromal, immune, and malignant epithelial cells play important roles in solid tumor progression and therapeutic response. CAFs play an integral part in the TME and can influence many aspects of carcinogenesis, including ECM remodeling, angiogenesis, cancer cell proliferation, invasion, inflammation, metabolic reprogramming, and metastasis [71].

Numerous clinical and pathological observations have established a clear relationship between chronic inflammation, fibrosis, and cancer [72]. Cancer development can be preceded or ensued by a fibrotic state that participate in multiple stages of tumorigenesis and metastasis [73].

In this review we have highlighted and described the multiple roles exerted by IGFBP-6 in all these processes, which suggest that it can be considered as a protein of interest also in the cancer development and progression. To corroborate this point, IGFBP-6 is directly related to immunological functions and inflammation activities in glioma, considered a potential therapeutic target for glioma immunotherapy [74]. Specifically, Zong Z. and colleagues recently demonstrated that IGFBP-6 is an unfavorable prognostic factor in glioma, affecting tumor malignancy with an expression positively correlated with the immunosuppressive response in glioma patients [75].

As already discussed, IGFBP-6 is highly expressed and exerts various actions in fibroblasts, a critical component of the TME during cancer progression [76] (Figure 2). Generally, IGF pathway is involved in fibroblast activation since IGFs/IGF-1R axis is linked to stromal fibroblast transition to CAFs. After c-Myc activation, the decrease of IGFBP-6 significantly enhances fibroblast activation and mobilization and increases the chemotactic potency of human primary breast cancer fibroblasts. As we have already highlighted, TGF-β has a suppressive role on IGFBP-6 [12,13]. Interestingly, it has been described that TGF-β works as a tumor suppressor during the early phases of the carcinogenesis, while it becomes a later promoter of it, as its overexpression may directly induce tumor metastasis by initiating events necessary for invasion [77,78]. These data confirm the dual role of TGF-β as both mediator and suppressor within the same process. Thus, IGFBP-6 is related to breast epithelial cell oncogenic activation, directly promoting TME remodeling and increasing tumor invasion [68]. Fibroblast growth factor-2 and IGFBP-6 are both activated in breast cancer by Vasohibin-2, an angiogenic factor [79]. IGFBP-6 also plays a crucial role in CAFs regulation, participating in epithelial–mesenchymal transition and contributing to glioma cell migration. It is also associated with CAFs infiltration in the stomach, colon, and rectal adenocarcinomas, underlying its important role in TME [80]. IGFBP-6 is significantly decreased in prostate cancer patients, compared to control subjects. Cancer cell proliferation and progression are facilitated by the formation of an immune cell infiltration microenvironment. During cancer progression, the TME also changes immune cell infiltration and IGFBP-6 correlates with B cells, CD4+T cells, CD8+T cells, neutrophils, macrophages, and DCs in patients with gastric cancer [80].

Interestingly, Wnt and Hedgehog (Hh) signaling pathways are also involved in IGFBP-6 regulation [5]. The Hh pathway is critical for cellular development, and the deregulation of this pathway is found in a relevant number of cancers [81]. IGFBP-6 levels are higher in prostate CAFs than in normal prostate fibroblasts, and its levels are regulated by Hh signaling [82,83]. Hh pathway is aberrantly activated in cancer with GLI Family Zinc Finger 1, maintaining cell survival by binding the promoter regions and facilitating the transcription of IGFBP-6 and Bcl-2 genes in colorectal carcinomas and pancreatic cancers [84,85]. In this complex scenario, our recent work demonstrated that IGFBP-6/SHH/Toll-like receptor4 axis is implicated in alterations of the primary myelofibrosis microenvironment and that IGFBP-6 may play a central role in activating the SHH pathway during the fibrotic process [58]. Bach L.A. considered that the IGFBP-6 increase after Hh pathway activation may be a counter-regulatory response to the regulation of IGF activity or it may represent an independent role for IGFBP-6 in the Hh pathway [5].

## 6. Conclusions

As IGFBP-6 is capable of IGF-II independent actions [17], including the regulation of proliferation, apoptosis, angiogenesis, cell migration, and fibrosis progression, we propose that this protein is a major player in immunity and in inflammation. Further studies will elucidate its possible and complex role in the regulation of immunological fibrosis, with particular relevance concerning its function in the microenvironment and in the transformation of fibroblasts into CAFs, contributing to cancer progression together with the fibrotic process.

## Figures and Tables

**Figure 1 ijms-23-04358-f001:**
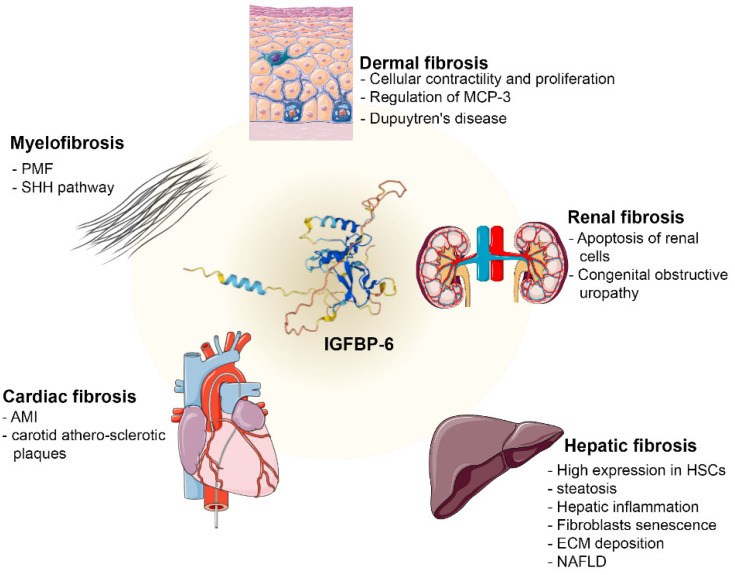
IGFBP-6 is involved in different types of fibrosis. Schematic representation of the main actions exerted by IGFBP-6 in the different types of fibrosis in which it is involved. (MCP-3: monocyte chemoattractant protein; HSCs: Hepatic Stellate Cells; ECM: Extracellular Matrix; NAFLD: Non-Alcoholic Fatty Liver Disease; PMF: Primary Myelofibrosis; SHH: Sonic Hedgehog; AMI: Acute Myocardial Infarction. Images credits: [47,48]).

**Figure 2 ijms-23-04358-f002:**
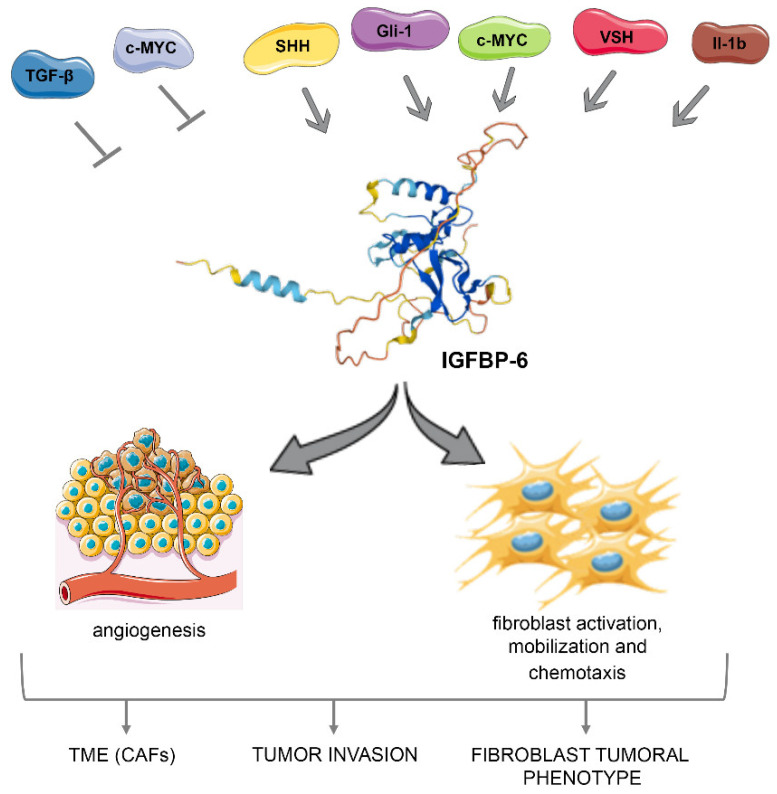
IGFBP-6 actions in fibroblasts and TME during cancer progression. Schematic representation of the cytokine, chemokine, and molecular networks involved in the regulation of IGFBP-6 with consequent involvement in controlling fibroblasts activation, tumor invasion and maintenance of the tumor microenvironment during cancer progression. (SHH: Sonic Hedgehog; VASH: Vasohibin; CAFS: Cancer Activated Fibroblasts. Grey arrowheads indicate an activation of IGFBP-6, while arrows without heads indicate an inactivation and de-regulation of IGFBP-6. Images credits: [47,48]).

**Table 1 ijms-23-04358-t001:** IGFBP-6 is involved in the progression of different kinds of fibrosis.

	Disease/Clinical Problem	Type of Sample	Number of Cases	Expression Level	Ref.
**IGFBP-6**	**Dermal Fibrosis**				
	Systemic sclerosis	Skin biopsy samples	Skin biopsy samples from the interscapular region of 5 Type 1 tight-skin (Tsk1) compared to 5 wild-type littermate mice at between 3 days and 12 weeks of age	UP (2-fold)	[49]
	Cellular contractility and proliferation in Dupuytren’s Disease (DD)	Primary Fibroblasts (PF) from diseased palmar fascia	3 affected patients compared to PF derived from the adjacent, phenotypically unaffected palmar fascia of the same patients	DOWN (*p* < 0.05)	[50]
	**Renal Fibrosis**				
	Proteomic study in Chronic Kidney Disease (CKD)	Human plasma samples	389 patients, 51% male (N = 201), with a median age of 64 years, and median body mass index of 25.3	DOWN (rho = −0.81, *p* = 1.0 × 10^−82^)	[51]
	End Stage Renal Disease (ERSD)	Human blood samples	16 patients with ESRD receiving hemodialysis (8 men and 8 women) compared to 19 control healthy subjects (10 men and 9 women).	UP (*p* < 0.05)	[52]
	Characterization of congenital obstructive uropathy	RNA from rats with congenital hydronephrosis	Total cellular RNA from obstructed (n = 16), contralateral (n = 10) compared to healthy control kidneys (n = 4)	UP (7.4 fold, *p* < 0.01)	[53]
	**Hepatic Fibrosis**				
	Disease severity and glycemia in nonalcoholic fatty liver disease	Human liver biopsy	61 patients with HIV-infection, ≥5% hepatic fat fraction	UP (Fibrosis stage *p* = 0.03; Steatosis grade *p* = 0.004; NAFLD activity score *p* = 0.003)	[54]
	Chronic hepatitis C (CHC)	Human blood samples	128 CHC patients and 123 controls	DOWN (*p* ≤ 0.001)	[55]
	**Cardiac Fibrosis**				
	Identification of new Acute myocardial infarction (AMI) biomarkers	Human plasma samples	10 AMI patients compared to 5 controls with no myocardial infarction	DOWN (Ratio = 0.70)	[56]
	Atherosclerotic carotid plaques	Gene Expression Omnibus (GEO) data sets and the European Bioinformatics Institute (EBI) database; Atherosclerotic plaques from patients with high-grade carotid artery stenosis	GSE41571 (5 ruptured and 6 stable carotid plaques); E-MTAB-2055 (25 ruptured and 22 stable plaques); GSE118481 (10 clinical unstable and 6 stable plaques); 52 patients (28 with stable plaques and 24 with unstable plaques)	DOWN (*p* < 0.0001)	[57]
	**Myelofibrosis**				
	Microenvironmental alteration of primary myelofibrosis	Healthy mesenchymal stem cells HS5	-	UP (*p* < 0.05)	[58]

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
