# Peer review of "IGFBP-6: At the Crossroads of Immunity, Tissue Repair and Fibrosis"

_ijms, 2022, doi:10.3390/ijms23084358_

Round 1

Reviewer 1 Report

The review article Liso, et al., describes the role of the Insulin-like Grow Factor-Binding Protein 6 (IGFBP6) in the immune system, tissue repair, and fibrosis. Although the topic might be interesting to the IJMS readers, the paper in its current form has several serious drawbacks that reduce its potential. While I am not a native speaker either, the English level in the manuscript is not sufficient. I would recommend the proofreading service or native English speaker to thoroughly check the paper, mostly syntax and grammar, otherwise, the information provided in the manuscript is hard to follow.

(Additionally, several times in the text there is a mixture of British or American English – tumor/tumour, signaling/signalling etc.)

Overall, the conclusions and a common link between the individual chapters or even sentences through this review are missing. For example, the review should be focused on IGFBP-6 (according to the title and abstract), however, chapter 3 tries to cover all of the IGFBPs. While reading it, there was an impression that the authors try to cover everything, but actually ended up saying too little about each. The table provided is not informative, the table summarizing all studies with IGFBP6 but with additional data would be more informative. I would highly recommend sticking to IGBP-6 only and discuss this in more depth and in detail – are these clinical studies/animal studies/in vitro studies (cell lines)? how many patients were included? by how much it was increased/decreased, what was the clinical problem? …. etc. This can be important to give the readers a more complex view on the role of IGFBP6 in fibrosis. Also, chapter 5 should be completely rewritten and actually, it should be moved to chapter 1, so the reader is introduced to the problematics as well as the importance of IGFBP6.

Major issues - In the 2nd chapter the statement “In this context, IGFBP-6 is an agonist of neutrophils’ functions like increased oxidative burst with ROS production, degranulation of primary granules, chemotaxis of T-cells and monocytes through the epithelial monolayer (16).” At the end of the paragraph authors provide the opposite argument: “More recently it was demonstrated that IGFBP-6 improves the mitochondrial fitness and redox, reducing the mitochondrial ROS production and modulating lactate metabolism and oxidative stress in a human breast cancer cell line (19)“. Due to the inconsistency in the effect of IGFBP-6 on ROS production, I would expect the author’s thoughts or explanation of these opposing arguments.

In the 5th chapter What does metazoans attack mean? The chapter is in my opinion very chaotic, providing no further conclusion. As mentioned above, the place of this chapter at the end of the text is questionable.

The summary or conclusion paragraph for each chapter is missing and the connection between the chapters is not clear. More importantly, the conclusion chapter at the end of the whole review including the future directions and pointing out the importance of this review is missing.

Other minor issues:

There are too many unexplained abbreviations starting with IGFBP6 as the first word in the abstract (same with SHH, DCs). Sometimes the abbreviations were explained a couple of paragraphs later and sometimes they were used just once, which makes them totally useless. On the other hand, some other abbreviations were not explained at all (CLT, TGF…. ).

(MCP-3, Tsk1, DD, NAS, NAFLD, NASH, CHC, HSC, BM, PMF, CKD, ERSD, GHRH, JAK2, etc….). Although some of them are notoriously known, they should be explained when they are mentioned for the first time. This can be omitted, should the term be used only once in the manuscript.  

As a whole, the topic has the potential to be interesting for the readers, nevertheless, in its current form, it is not suitable for publication. Concise overwrite and English proofreading is necessary prior to resubmission.

Author Response

Response to Reviewer 1 Comments

Q: I would recommend the proofreading service or native English speaker to thoroughly check the paper, mostly syntax and grammar, otherwise, the information provided in the manuscript is hard to follow.

A: We have checked the syntax and the grammar and we have standardized the language of the manuscript, using American English throughout the text.

Q: Overall, the conclusions and a common link between the individual chapters or even sentences through this review are missing. For example, the review should be focused on IGFBP-6 (according to the title and abstract), however, chapter 3 tries to cover all of the IGFBPs.

A: We thank the Reviewer for this comment and for giving us the possibility to improve the manuscript. We have inserted conclusions and a common link at the end of each paragraph.

Q: The table provided is not informative, the table summarizing all studies with IGFBP6 but with additional data would be more informative. I would highly recommend sticking to IGBP-6 only and discuss this in more depth and in detail – are these clinical studies/animal studies/in vitro studies (cell lines)? how many patients were included? by how much it was increased/decreased, what was the clinical problem?

A: As suggested by the Reviewer, we completely modified Table 1, replacing it with a new one in which we have excluded all the other IGFBPs, focusing on the IGFBP-6 actions in fibrosis, and entering more detailed information.

Q: Also, chapter 5 should be completely rewritten and actually, it should be moved to chapter 1, so the reader is introduced to the problematics as well as the importance of IGFBP6.

A: Thanks for this suggestion. We have modified the chapter, accordingly.

Q: In the 2nd chapter the statement “In this context, IGFBP-6 is an agonist of neutrophils’ functions like increased oxidative burst with ROS production, degranulation of primary granules, chemotaxis of T-cells and monocytes through the epithelial monolayer (16).” At the end of the paragraph authors provide the opposite argument: “More recently it was demonstrated that IGFBP-6 improves the mitochondrial fitness and redox, reducing the mitochondrial ROS production and modulating lactate metabolism and oxidative stress in a human breast cancer cell line (19)“. Due to the inconsistency in the effect of IGFBP-6 on ROS production, I would expect the author’s thoughts or explanation of these opposing arguments.

A: We thank the Reviewer for this important comment. IGFBP-6 is a secreted protein that performs various immunological functions, as previously stated. Although several studies have highlighted its chemotactic and pro-inflammatory role, further evidence demonstrates its possible double face role, also placing it among the anti-inflammatory proteins. This type of regulation has been amply demonstrated on various proteins such as Interferon-β (Bolivar S. et al., 2018) or Interleukin-6 (Scheller J. et al., 2011) that exert both pro-inflammatory and anti-inflammatory effects. Further studies are needed to clarify what is the IGFBP-6 predominant role or whether indeed the protein can play both roles. We have made this point clearer, as requested.

Q: In the 5th chapter What does metazoans attack mean? The chapter is in my opinion very chaotic, providing no further conclusion. As mentioned above, the place of this chapter at the end of the text is questionable.

A: We agree with the Reviewer that this issue is complex and it requires to be clarified. Accordingly, we have now divided the 5th paragraph into two parts by inserting the first more general part immediately after the 1st paragraph (now 2nd paragraph), as suggested, and leaving the second part, more specific and closer to the conclusions, within the 5th paragraph. The text has been completely revised and corrected both in form and in English.

Q: The summary or conclusion paragraph for each chapter is missing and the connection between the chapters is not clear. More importantly, the conclusion chapter at the end of the whole review including the future directions and pointing out the importance of this review is missing.

A: According to the Reviewer’s suggestion, conclusions have been inserted at the end of each paragraph to better link it to the next paragraph. In addition, a “Conclusions” section was inserted at the end of the entire manuscript (6th paragraph), as requested by the Reviewer.

Minor critisims:

Q: There are too many unexplained abbreviations starting with IGFBP6 as the first word in the abstract (same with SHH, DCs). Sometimes the abbreviations were explained a couple of paragraphs later and sometimes they were used just once, which makes them totally useless. On the other hand, some other abbreviations were not explained at all (CLT, TGF…. ).

(MCP-3, Tsk1, DD, NAS, NAFLD, NASH, CHC, HSC, BM, PMF, CKD, ERSD, GHRH, JAK2, etc….). Although some of them are notoriously known, they should be explained when they are mentioned for the first time. This can be omitted, should the term be used only once in the manuscript.  

A: We have revised all abbreviations and modified those which were not correctly indicated, also others have been eliminated, as requested, when they were used only once in the text.

Reviewer 2 Report

The present manuscript by Liso et al provides an updated overview of IGFBP-6 biology. The review focuses on the roles of this binding protein in immunologic and fibrotic processes. In general terms, the review is highly comprehensive though sometimes specific paragraphs are not very focused. The figures are very illustrative.

Specific points:

1.Abstract, first sentence, authors chose to emphasize right at the beginning of the paper the role of IGFBP-6 in response to hyperthermia. This is kind of strange given that hyperthermia is not the focus of the paper.

  1. Page 2, second paragraph, sentence starting “Several studies will be needed…” is grammatically wrong. Please fix.

  1. Page 2, Section 2, first paragraph, sentence starting “Since stressful conditions …” is very long and confusing. Please divide into two sentences.

  1. Page 5, bottom of Table 1, sentence starting “ The IGFs system is an important complex ” is out of context. Please remove.

  1. Page 5, bottom paragraph, sentence starting “IGFBP-6 is high expressed in fibroblasts …”, should be “highly expressed”. Also, should be “tissue maintenance” (not maintaining).

  1. Page 7, Section 4.2, should be “Wang AND colleagues…”.

  1. Page 7, Section 4.3, first sentence, should be “constitutively produce high QUANTITIES of IGFBPs”.

  1. In the same paragraph, sentence starting “Interestingly, HSC isolated …” is extremely confusing. Please rephrase.

  1. Page 7, Section 4.3, bottom of second paragraph, sentence starting “Moreover, IGFBP-6 was downregulated …”, should be “it may PARTICIPATE”.

  1. Page 7, Section 4.3, third paragraph, no need to spell out IGF binding proteins.

  1. Page 8, Section 4.5, second paragraph, should be “implicated in the pathogenesis of primary…”

  1. Page 8, Section 5, second paragraph, sentence starting “Often, the Th2 actions…”, should be “promote” (not promotes).

  1. Page 9, line 5, should be TNF-alpha.

  1. Page 9, third paragraph, sentence starting “An American study …”. Please, be more specific. Cite author, not continent of origin.

  1. Page 10, bottom of first paragraph, sentence starting “Carcinomas modify…” is grammatically wrong. Please rephrase.

  1. It is necessary to include a Conclusions section.

  1. Reference 60 is incomplete.

Author Response

Response to Reviewer 2 Comments

Q: Abstract, first sentence, authors chose to emphasize right at the beginning of the paper the role of IGFBP-6 in response to hyperthermia. This is kind of strange given that hyperthermia is not the focus of the paper.

A: According to the Reviewer’s suggestion, we have modified the sentence at the beginning of the abstract. 

Q: Page 2, second paragraph, sentence starting “Several studies will be needed…” is grammatically wrong. Please fix.

A: We have modified the sentence.

Q: Page 2, Section 2, first paragraph, sentence starting “Since stressful conditions …” is very long and confusing. Please divide into two sentences.

A: As suggested by the Reviewer, we have divided the sentence into two parts, to make the text clearer.

Q: Page 5, bottom of Table 1, sentence starting “The IGFs system is an important complex ” is out of context. Please remove.

A: We have removed the sentence, as suggested.

Q: Page 5, bottom paragraph, sentence starting “IGFBP-6 is high expressed in fibroblasts …”, should be “highly expressed”. Also, should be “tissue maintenance” (not maintaining).

Page 7, Section 4.2, should be “Wang AND colleagues…”.

Page 7, Section 4.3, first sentence, should be “constitutively produce high QUANTITIES of IGFBPs”.

A: We have modified the text according to the Reviewer’s comments.

Q: In the same paragraph, sentence starting “Interestingly, HSC isolated …” is extremely confusing. Please rephrase.

A: We have corrected the sentence in section 4.3 (section 4.1.3 in the revised version of the manuscript) to make it clearer.

Q: Page 7, Section 4.3, bottom of second paragraph, sentence starting “Moreover, IGFBP-6 was downregulated …”, should be “it may PARTICIPATE”.

Page 7, Section 4.3, third paragraph, no need to spell out IGF binding proteins.

Page 8, Section 4.5, second paragraph, should be “implicated in the pathogenesis of primary…”

Page 8, Section 5, second paragraph, sentence starting “Often, the Th2 actions…”, should be “promote” (not promotes).

Page 9, line 5, should be TNF-alpha.

Page 9, third paragraph, sentence starting “An American study …”. Please, be more specific. Cite author, not continent of origin.

A: We thank the Reviewer for all these suggestions. We have modified the text according to the Reviewer comments. Please note that the changes made within paragraph 5th are now in the current 2nd paragraph, as a result of the changes made in the text.

Q: Page 10, bottom of first paragraph, sentence starting “Carcinomas modify…” is grammatically wrong. Please rephrase.

A: We really thank the Reviewer for this comment. We corrected the sentence in section 5 to make it correct.

Q: It is necessary to include a Conclusions section.

A: As suggested by the Reviewer, we have included “Conclusions” in the 6th section, at the end of the manuscript.

Q: Reference 60 is incomplete.

A: Since we have rewritten some parts of the text and the table, this reference has been removed from the text.

Round 2

Reviewer 2 Report

Page 4, third paragraph, should be 'IGFBP-6' (not IFBP-6)